# Green Extraction of Graphene from Natural Mineral Shungite

**DOI:** 10.3390/nano12244356

**Published:** 2022-12-07

**Authors:** Anastasia Novikova, Alina Karabchevsky

**Affiliations:** School of Electrical and Computer Engineering, Ben-Gurion University of the Negev, Beer-Sheva 8410501, Israel

**Keywords:** green graphene, mineral, shungite, produce graphene, sonication

## Abstract

Conventional fabrication methods to produce graphene are cumbersome, expensive, and not ecologically friendly. This is due to the fact that the processing of a large volume of raw materials requires large amounts of acids and alkalis which, in turn, require special disposal. Therefore, it is necessary to develop new technologies or to refine existing ones for the production of graphene—and to create new, ecologically-safe and effective methods. Here, we utilized physical sonication to extract graphene films from natural mineral shungite rock. From our study of the structure of shungite by Raman spectrometry and X-ray phase analysis, we found that shungite is characterized by graphite-like mineral structures. Transmission electron microscopy images obtained from the processed material revealed graphene films—with surfaces as small as 200 nanometers long and several layers wide. Our green method of fabicating graphene can be widely used in a variety of fields, from electro-optics to ecology, to list a few.

## 1. Introduction

At the moment, graphene is a material that is being actively researched as it finds wide application in various industries [1,2,3,4,5]. Graphene has high thermal conductivity, electrical conductivity, flexibility, elasticity, mobility, and transparency [6,7,8,9,10,11]. Graphene layers also have a high specific surface area and antimicrobial properties [12,13]. Graphene is being used in various applications such as electronics, optics, and biotechnology [14,15], and has the potential to be used in other applications and industries in the future [2,4,16,17,18,19,20,21,22,23,24].

Graphene films are produced by the following methods [25,26,27]: the chemical cleavage method (treatment of graphite with a mixture of sulfuric and nitric acids [28,29,30,31], reduction of monolayer films of graphene oxide [32]), mechanical cleavage method (graphite is placed between sticky tapes and graphene layers are split off) [33], radio-frequency plasma chemical deposition from a gaseous medium [34] and growth of graphene films at high temperatures [35] and pressure [36]. For more information about graphene production methods, see Table 1.

Each of the presented methods has its pros and cons; during chemical processing, there is a need to wash the resulting graphene of the substances with which the material reacted, which increases the time to obtain graphene. During the mechanical processing (exfoliation) of graphite, large sheets of graphene are obtained; however, in this case, a large amount of tape is needed, which can also increase the amount of side debris, and the loss of some graphene on the tape. When grown on a substrate, graphene layers have a smaller film size than when machined. High energy consumption is required during plasma treatment. We chose ultrasound treatment because when using mechanical methods, films with large surfaces are formed. This method includes only two stages and is easy-to-use. In addition, it is environmentally friendly since it uses an aqueous medium without the addition of surfactants.

Here, we used shungite in the sonication which consists of hybrid structures of sp1+sp2+sp3 forms of hybridized carbon atoms (sp1 is carbyne, sp2 is graphene, and sp3 is graphite forms). Mostly, sp2 + sp3 forms structures are present in a small fraction of sp1 [45]. It is known from the literature that graphene can be obtained from shungite [46] since the mineral contains an sp2 hybridized form of carbon; the the graphite form can also be converted into graphene, which will increase the percentage of output material from raw materials [47,48,49].

Here, we report the production of graphene from shungite rock, as illustrated in Figure 1. Shungite and shungite-bearing rocks are included as part of a large group of Precambrian carbon-bearing rocks [50,51,52,53]. The main deposits of Shungite are located in Karelia, Russia. Shungite rocks differ in physicochemical and chemical properties, and mineralogical composition of diverse forms, besides varying in formation times; the genesis and material composition of the ash content; isotopic composition; and the structural state of shungite carbon [54,55,56]. Shungite rocks contain carbon in amorphous form (from 5 to 99% depending on the species), minerals (quartz, feldspar, aluminosilicates, carbonates, pyrites), small amounts of bitumen-like organics and water [57].

## 2. Experimental Section

### 2.1. Study of the Physical and Chemical Properties of Shungite

Shungite sample from the Zazhoginsky field were retrieved during geological processes. We investigated the surface of the shungite, the presence of mineral inclusions, and the composition of the elements. The studies were carried out using a scanning electron microscope (SEM) with a low vacuum and a Quanta 200 SEM tungsten electron source. The Quanta SEM electron microscope was also equipped with an energy-dispersive X-ray spectroscopy (EDS) system for elemental analysis. For the analysis of shungite, we also used the X-ray method. The studies were carried out using a PANalytical Empyrean multipurpose diffractometer for the analysis of powder and solid substances, nanomaterials, thin films, and suspensions.

### 2.2. Extraction of Graphene from Shungite

To test the technique, we used shungite from the Zazhoginsky deposit of group I (group I—more than 96% carbon). Before producing graphene, we broke up the shungite samples. A FRITSCH grinding mill 6 was used to grind shungite samples. Grinding time is 10 min at a room temperature of 25 ${}^{{}^{\circ}}$C. Samples were then cleaned of visible contaminants and washed in distilled water for 5 min at room temperature of 25 ${}^{{}^{\circ}}$C. Samples of graphene layers were prepared by the sonication method using a digital ultrasonic cleaner R Technology under normal conditions (temperature 25 ${}^{{}^{\circ}}$C). One gram of shungite was placed in a 50 mL plastic tube, 25 mL of distilled water was added and the sealed tube was placed in a digital ultrasonic cleaner R Technology for 2 h. We used the transmission electron microscope (TEM) JEOL JEM 2100F to evaluate and characterize the samples obtained. To assess the structural features, we studied the Raman spectra using a LabRam HR Evolution Horiba Raman spectrometer with an excitation range of 325–785 nm with an ultra-low frequency (ULF) module, which allows one to determine the characteristics of the sample at very low frequencies (532 nm). To study the composition of the elements, we used X-ray photoelectron spectroscopy (XPS). The studies were carried out using the ESCALAB 250, a multifunctional instrument which includes surface-sensitive XPS and Auger electron spectroscopy (AES) analysis methods.

## 3. Results

We checked the shungite surface using a scanning electron microscope (SEM) to determine the type of shungite. Figure 2 shows scanning electron microscope images of the surface.

We observed the heterogeneous and stepped surface of shungite; the surface-shell spalls characteristic of shungite, as can be seen from Figure 2a,b, has small numbers of micropores, as can be seen in (Figure 2c,d). The pores are unevenly distributed over the entire surface and are present only in spongy mineral inclusions (Figure 2c,d). The mineral contains inclusions of different types, and individualized grains and spongy inclusions of micro- and nanometer size, different in shape. In addition, mineral inclusions can have a different elemental composition.

After detecting inclusions of various shapes on the surface, we investigated the chemical composition of shungite using energy-dispersive X-ray spectroscopy (EDS). We selected random regions in the shungite and selected several areas in each region with the following dimensions: S1=300 nm${}^{2}$, S2=270 nm${}^{2}$, S21=7200 nm${}^{2}$ and S22=900 nm${}^{2}$ (shown in Figure 3a,d). The surfaces and the spectra of the selected areas are shown in Figure 3.

We investigated two areas in each chosen region, as shown in Figure 3a,d. The EDS spectra of areas 1 (Figure 3b) and 21 (Figure 3e) show that the completely dark sections of the natural mineral are 100% carbon. These data confirm that this mineral can be classified as shungite of group I, which has a large concentration of carbon. The EDS spectra of regions 2 (Figure 3c) and 22 (Figure 3f) show that, in addition to carbon, there are impurities (low concentrations) of oxygen, silicon, aluminum, nickel, iron, and vanadium, verifying that the sample is shungite. Concentrations of other substances in comparison with the total volume are very small; in further studies, we physically took parts of the samples without visible contamination.

Shungite is also characterized by graphite and graphene shapes, respectively, which have specific absorption peaks at specific wavelengths.

Using the method of X-ray spectrometry and Raman spectroscopy, we studied how carbon binds to other elements. The data obtained are shown in Figure 4.

Figure 4 shows Raman spectra of natural shungite in different areas. We analyzed peaks detected in two spectral regions: 1100–1800 cm${}^{-1}$ and 2550–3100 cm${}^{-1}$. The peaks that appear in the region of 1100–1800 cm${}^{-1}$ are of the first order and can be associated with the peaks of graphite forms of carbon. The peaks that appear in the region of 2550–3100 cm${}^{-1}$ are of the second order, also associated with graphite [58]. The band at 1600 cm${}^{-1}$ is the G-band, which appears due to the tangential stretching vibrations of carbon atoms in the hexagons of graphene planes. It appears in the Raman spectra of carbon materials with sp2 bonds; in our case, graphene. The band located at 1330 cm${}^{-1}$ is the D-band, which appears in the presence of diamond-like sp3 -bonds. In our case, it corresponds to the amorphous structural state of carbon (graphite). The location of those peaks indicates that the main part of amorphous carbon in shungite rock is graphite. The ratio of the intensities of the D (graphite) and G (graphene) bands is traditionally used to assess the degree of the ordering of carbon materials. In our case, the calculation was performed according to the height of the observed peaks (absolute maxima) of the intensity for D1 = 1343 cm${}^{-1}$, D2 = 1345 cm${}^{-1}$, D3 = 1345 cm${}^{-1}$, and for G1 = 1584 cm${}^{-1}$, G2 = 1585 cm${}^{-1}$, G3 = 1590 cm${}^{-1}$. From the ratio of the intensities of the D and G bands (Figure 4a), it can be seen that the structure of shungite is more disordered. The second-order peaks also differ in range, being offset as shown in (Figure 4a), representing three different peaks: 2550–3100 cm${}^{-1}$, D1${}^{\prime \prime }$ = 2657 cm${}^{-1}$, D2${}^{\prime \prime }$ = 2657 cm${}^{-1}$, D3${}^{\prime \prime }$ = 2654 cm${}^{-1}$, G1${}^{\prime \prime }$ = 2913 cm${}^{-1}$, G2${}^{\prime \prime }$ = 2938 cm${}^{-1}$, G3${}^{\prime \prime }$ = 2930 cm${}^{-1}$, which also indicates the disorder of the structure of shungite. Figure 4b shows the Raman spectrum of a sonicated shungite sample. An analysis was performed on the peaks recorded in two spectral regions: 1100–1800 cm${}^{-1}$ and 2550–3100 cm${}^{-1}$. From the obtained Raman spectra, we can conclude that the spectra of the treated shungite are different. There are no evident second-order peaks and the intensity of the D and G lines changed. Peak position is D1 = 1359.82 cm${}^{-1}$, D2 = 1359.82 cm${}^{-1}$, G1 = 1639.89 cm${}^{-1}$, G2 = 1619.98 cm${}^{-1}$. The ratio of the peaks also changed, which indicates that structural changes have occurred in the samples. In untreated shungite, the D peak related to sp3 bonds (graphite) prevailed; in the treated sample, it greatly decreased. The G line related to sp2 bonds (graphene) expanded. The peaks moved to the right. The obtained Raman spectra data confirm the results obtained by the XPS and TEM data.

To determine the numbers of sp2 and sp3 carbon bonds, we studied the samples using X-ray photoelectron spectrometry. Data are presented in Table 2 and in Figure 5.

The XPS spectra of shungite after deconvolution is shown in Figure 5a,c and summarized in Table 2. Shungite has four peaks, at 284.32 eV, 284.55 eV, 287.31 eV, and 285.85 eV, bonds corresponding to sp3 hybridized carbon (C-C), sp2 hybridized carbon (C=C), carbonyl group (C=O) and hydroxyl groups (C-OH), respectively [59]. In addition, C=O peaks are presented at 532.34 eV and C-OH at 533.05 eV [60]. Shungite samples before processing contain 49.73% hybridized carbon, next is hydroxyl groups at 19.88%; most likely, this is due to the presence of water in the samples. Quantitative analysis of carbon and oxygen is presented in the Appendix A. Data confirms previously obtained information from X-ray and Raman spectroscopies.

After examining the shungite samples, we processed the samples via a mechanical method—sonication. Since large volumes of samples can be used with this method, there is no secondary contamination and there is no need to carry out subsequent sample processing.

The XPS spectra of sonicated shungite after deconvolution is shown in Figure 5b,d and summarized in Table 3. Table 3 shows the peak of the sp2 form of carbon, located at 284.58 eV, which is attributed to carbon. The full width at half maximum of the spectral line (FWHM) was 1.44 eV. The peaks are broadened as compared to the peaks before the treatment, which means that there is a decrease in the size of the coherent scattering region along the c-axis in the stacked graphene layers, which is confirmed in the literature [60,61]. Comparing the percentage of sp2 and sp3 forms, we can conclude that there are more sp2 forms; the percentage of sp3 forms has increased by 3.04% and the percentage of C-H has decreased by 8.89%, while the C=O bonds have totally disappeared which is confirmed by Figure 5d: elements identifier and quantitative analysis shows the decrease in the number of oxygen bonds, their number being reduced by 3.79%. In addition, C-O bonds appeared, which suggests that during sonication, one of the C=O bonds was destroyed. The initial concentration of the graphene form was 49.73%; after treatment, the concentration increased to 53.63%.

Figure 6a–c shows TEM images of the dispersed shungite sample, showing that the sonication process crushes the particles to form thin films with high specific surfaces. For a sample with a carbon concentration of 98%, we see that the particles have collapsed into thin monolayers. Knowing the characteristics of shungite, we can say that the graphene form is depicted in the figure. We also see that larger particles are present, which means that for this material, it is necessary either to increase the processing time or to separate thin layers from particles and further process the material. Figure 6d shows a fast Fourier transform pattern, a symmetrical hexagonal pattern which is characteristic of primordial graphene [62,63].

Figure 7 shows X-ray spectra of shungite before and after sonication. For sample (Figure 7a), wide spectra at 26.5${}^{{}^{\circ}}$ refer to carbon in its amorphous form and, more specifically, to graphite-like carbon, as well as a secondary peak at 42${}^{{}^{\circ}}$, 44${}^{{}^{\circ}}$ corresponding to a hexagonal graphite structure. Here, 26.5${}^{{}^{\circ}}$ is the angle of the basic plane of centered graphene, which corresponds, according to the literature data [64,65], to a lattice step of 0.335 nm. (Figure 7b) shows X-ray spectra after sonication. The peaks’ positions are located at 26.5${}^{{}^{\circ}}$, 42${}^{{}^{\circ}}$, and 44${}^{{}^{\circ}}$ and correspond to the structure of hexagonal graphene. The peak is 26.5${}^{{}^{\circ}}$, widened compared to the peak before treatment, which means that the carbon peak can be interpreted as a decrease in the size of the coherent scattering region along the c-axis in the stacked graphene layers. The peak is 44${}^{{}^{\circ}}$, more intense than that of shungite before sonication, which means that the size of the gescogonal structure was increased.

## 4. Discussion

We considered the production of graphene films from natural mineral shungite using a mechanical method—low-temperature (25 ${}^{{}^{\circ}}$C) sonication without the need for surfactants to stabilize the films. This is a promising technique for producing industrial volumes of graphene. Films with high concentrations of shungite in water were also obtained for the first time. The method we report is efficient and easy to use, does not require additional costs for treatment with chemical reagents, does not imply secondary contamination of the environment with surfactants, and does not imply the use of high temperatures, thus creating new prospects for the use of green-graphene.

The samples obtained are in the form of films with a surface length of 200 nm with a structure of hexagonal-centered graphene with a lattice pitch of 0.335 nm, which is confirmed by X-ray data and RAMAN spectra. The outcomes of this research will open the door to a new and environmentally safe method of graphene production. Green fabrication of graphene can be used in a variety of fields, from electronics (electrodes) and optics to biotechnology (biosensors, drug delivery, bio-medicine), energy (supercapacitors, solar cells, energy harvesting), and ecology (sorbents for wastewater, air purification).

## Figures and Tables

**Figure 1 nanomaterials-12-04356-f001:**
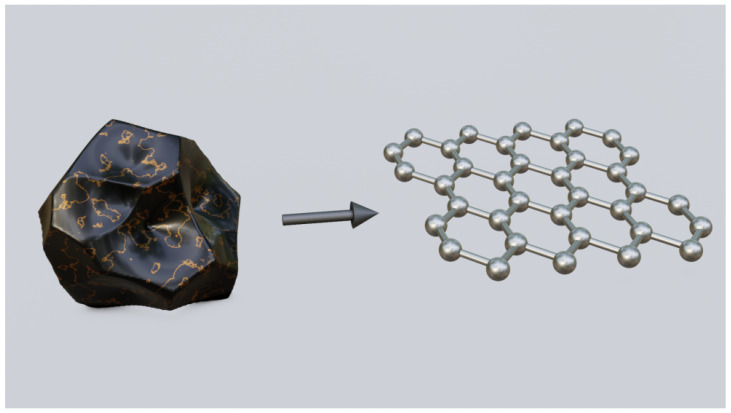
Illustration of Shungite rock and the graphene layer.

**Figure 2 nanomaterials-12-04356-f002:**
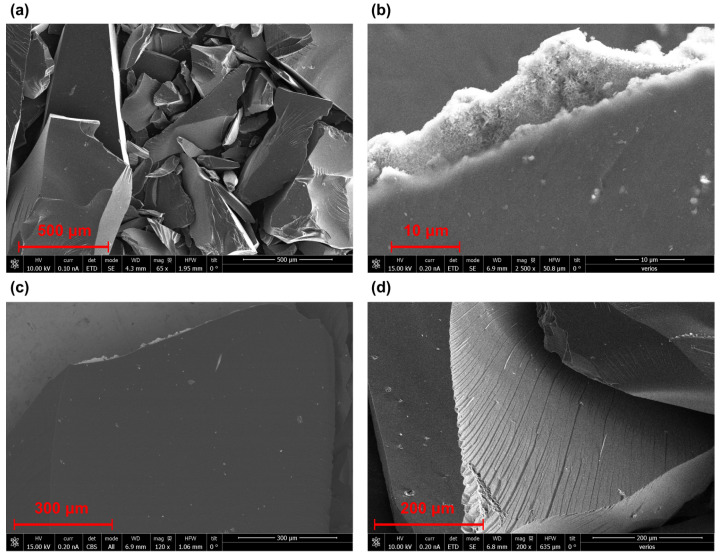
Scanning electron microscopy images of shungite particles: (**a**) general view of particles at a magnification of ×65, (**b**) shungite surfaces with chemical inclusions at a magnification of ×2500, (**c**) shungite surfaces with chemical inclusions at a magnification of ×120 and (**d**) shungite surface with characteristic chips particles at a magnification of ×200.

**Figure 3 nanomaterials-12-04356-f003:**
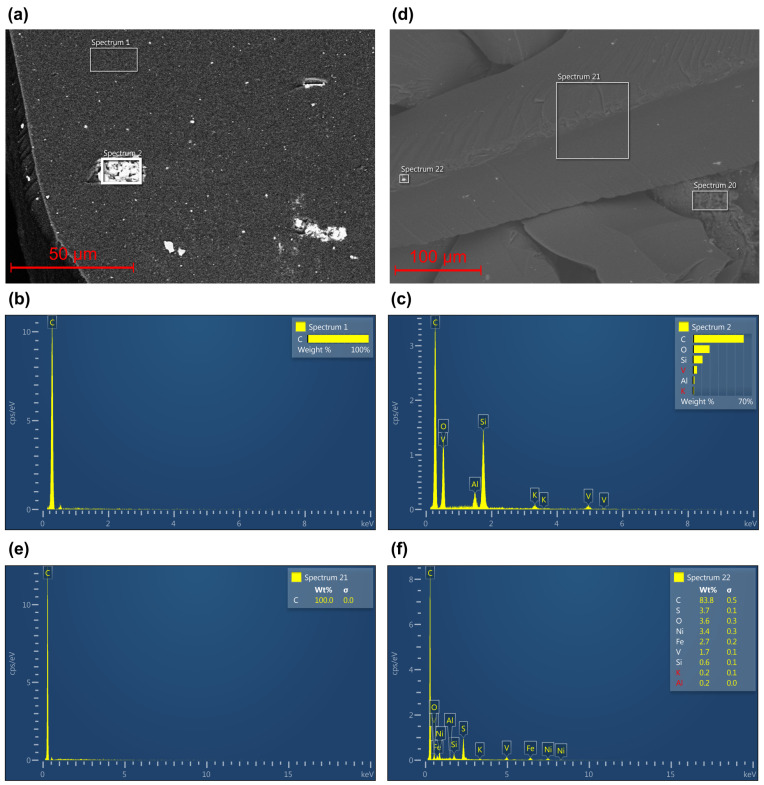
Scanning electron microscopy with EDS image of shungite particles: (**a**) first-selected sample study areas, (**b**) chemical composition of area spectrum 1, (**c**) chemical composition of area spectrum 2, (**d**) second-selected sample study areas, (**e**) chemical composition of area spectrum 21 and (**f**) chemical composition of area spectrum 22.

**Figure 4 nanomaterials-12-04356-f004:**
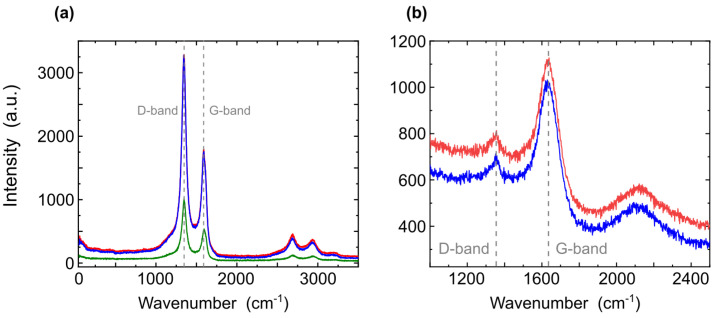
Raman spectra of natural shungite (**a**) before and (**b**) after sonication.

**Figure 5 nanomaterials-12-04356-f005:**
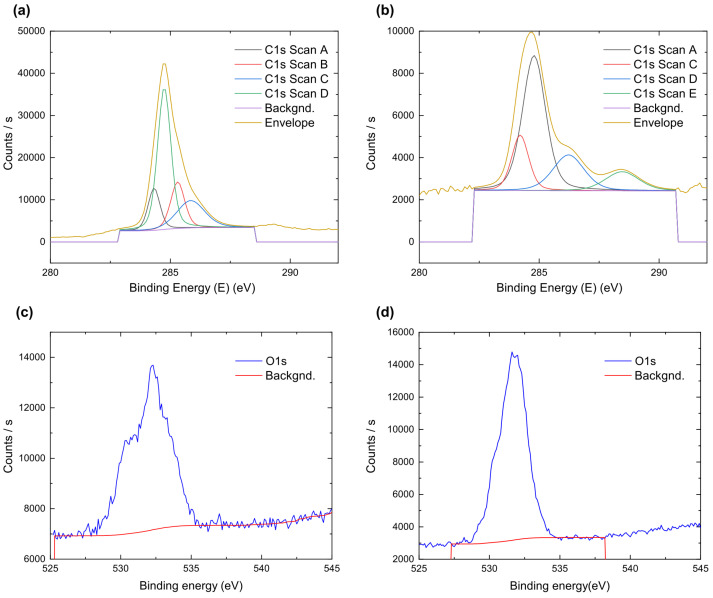
XPS spectra of (**a**) carbon peaks before sonication, (**b**) carbon peaks after sonication, (**c**) oxygen peaks before sonication, and (**d**) oxygen peaks after sonication.

**Figure 6 nanomaterials-12-04356-f006:**
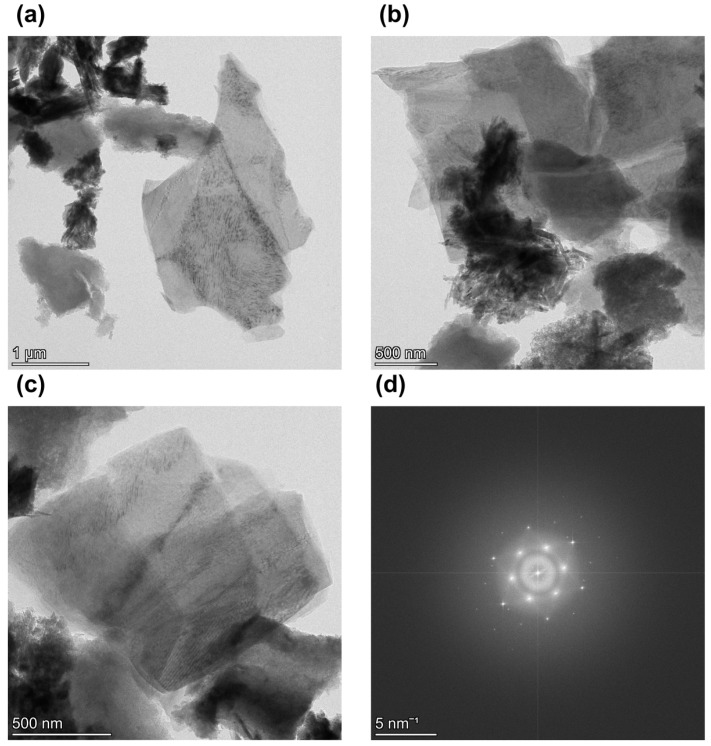
TEM images of sonicated shungite sample at (**a**) at ×13,500 magnification and (**b**) at ×35,000 magnification, (**c**) at ×22,000 magnification, and (**d**) fast Fourier transform.

**Figure 7 nanomaterials-12-04356-f007:**
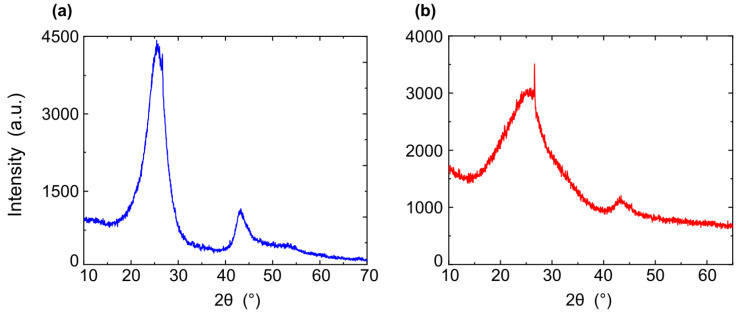
X-ray spectra of shungite (**a**) sample before sonication, (**b**) sample after sonication.

**Table 1 nanomaterials-12-04356-t001:** The main methods of obtaining graphene.

Method	Time	Number of Stages	Materials Used	Ref.
Chemical methods	12–24 h	More than 3	Acids and alkalis	[37,38]
Reduction of monolayer	24 h	More than 3	Acids and alkalis	[39,40]
films of GO				
Mechanical methods	4–6 h	More than 5	Alcohol	[41]
Plasma methods	1–7 min	1	N-silicon	[42,43]
Growth methods	10–12 h	1	SiC, Ge, acids	[44]
Sonication	2 h	1	-	This work

**Table 2 nanomaterials-12-04356-t002:** Elements identifier and quantitative assessment of shungite samples before sonication.

Name	Peak	FWHM	Area	Atomic
	(BE)	(eV)	(CPS·eV)	(%)
C1S scan A sp3 C-C	284.32	0.59	7059.83	13.25
C1S scan D sp2 C=C	284.55	0.66	26,502.12	49.73
C1S scan B C=O	287.31	0.69	9135.39	17.14
C1S scan C C-OH	285.85	1.39	10,594.59	19.88

**Table 3 nanomaterials-12-04356-t003:** Elements identifier and quantitative assessment of shungite samples after sonication.

Name	Peak	FWHM	Area	Atomic
	(BE)	(eV)	(CPS·eV)	(%)
C1S scan C sp3 C-C	284.21	0.84	1645.01	16.29
C1S scan A sp2 C=C	284.58	1.44	8709.01	53.63
C1S scan E C-O	286.32	1.55	3098.99	19.09
C1S scan D C-OH	285.86	1.66	1783.84	10.99

## Data Availability

Data underlying the results presented in this paper are not publicly available at this time but may be obtained from the authors upon reasonable request.

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
