# Peer review of "Green Extraction of Graphene from Natural Mineral Shungite"

_nanomaterials, 2022, doi:10.3390/nano12244356_

Round 1
Reviewer 1 Report
This work describes the process of extraction of graphene from natural mineral shungite by means of sonication.
In my opinion, the quality of the manuscript could be interesting in terms of perspectives and readibility, but not suitable in its actual form for publication on Nanomaterials, because of the lack of attention in the editing (see the list of comments below) and poor XPS data interpretation.
Here my comments:
1) Why are line numbers missing? They would have been useful in the reviewing process.
2) More references are necessary in the introduction (first 4 lines)
3) More info on shungite should be given in the final part of introduction
4) Last sentences of the introduction:
"Shungite and shungite-bearing rocks are included part of in a large group of": too many prepositions.
5) Fig. 1: size can be reduced, the presented image is not so significant
6) Fig. 2: images b and d are inverted respect to the description in the caption
7) Were the samples heated or cleaned before XPS analysis? Water on the surface could alteratr the obtained results
8) XPS spectra are necessary, not only quantitative analysis in Table 1 and 2
9) The term percent should be replaced with the symbol % in the whole manuscript.
10) XPS analysis: C=O bonding hypothesis must be also supported by quantitative analysis of the relative intensities of C 1s and O 1s related components, after the deconvolution of the peaks.
11) The sentence at page 5 "The full width at half maximum of the spectral line (FWHM) was 1.44 eV" is not clear: this parameter doesn't represent the resolution of the spectrometer.
It is not clear to me why the deconvolution of the photoeletric line in different peaks should give peaks with different FWHM: there are no physical reason for that. Was the deconvolution procedure adequate?
12) Last sentence of page 5: "sample with a carbon concentration of 98". Probably it is a percentage, but it must be indicated.
13) Fig. 5d: in which point of Fig 5c the diffraction pattern was collected? This information must be clarified.
14) Fig. 6: the 2 graphs presented in the figure look identical, no differences at all: from my point of view, this is not consonant with previous results, where significant differences before and after the sonication process are evident.
15) Discussion paragraph, second line: "a mechanical method - low-temperature (25 degree)" must be replaced with 25 °C.
16) Impersonal verbs and passive forms have to be used in the whole manuscript, instead of the personal "we".
Author Response
Good day, please see the attachment.
Kind regards,
Anastasia Novikova

Reviewer 2 Report
In this paper, the authors used physical ultrasound to extract graphene films from natural mineral syenite. Through Raman spectroscopy and X-ray phase analysis of the structure of the cis feldspar, the authors found that the cis feldspar has the characteristics of the graphite like mineral structure. The green manufacturing methods of graphene reported by the authors can be widely used in various fields, from electronics (electrodes) and optics, to biotechnology (biosensors) and ecology (adsorbents are used for wastewater and air purification). I believe that publication of the manuscript may be considered only after the following issues have been resolved.
1. In order to better highlight the advantages of this work, the author needs to provide a table to compare related work.
2. The summary needs to be rewritten. The application background is mainly in the introduction, which cannot be expanded too much in the abstract. The summary part needs to focus on the work of this paper.
3. There are few keywords in this article, and one or two related keywords need to be added.
4. The text information in Figure 3 is not clear enough, and the author needs to make adjustments.
5. The introduction can be improved. The articles related to the some applications of graphene materials should be added such as Sensors 2022, 22, 6483; ACS Sustain. Chem. Eng. 2015, 3, 1677–1685; RSC Adv. 2022, 12, 7821–7829; Talanta 2015, 134, 435–442.
6. Please check the grammar and spelling mistakes of the whole manuscript.
Author Response
Good day, please see the attachment
Kind regards,
Anastasia Novikova

Round 2
Reviewer 1 Report
The authors replied to some of the comments and made some changes in paper, but still, in my opinion, different points must be discussed and improved before accepting for publication on Nanomaterials.
1) Still lack of attention in the editing of the presented article:
- Table 1 is cutted,
- Lines 181 and followings: the term degrees is inappropriate, while the symbol "°" must be used
- in my opinion, in a high level scientific article the use of the passive form and/or infinitive verbs is necessary.
2) The authors replied: "The sample were not processed thermally or chemically before XPS analysis".
This is a critical point, because the surface composition is surely altered respect to the bulk if not treated before XPS analysis, and the results obtained are not truly significant: as a proof of my position, the authors disprove themselves in the manuscript presenting in Fig. 3 different element in the composition of the samples, and then stating that with XPS they can only see "C, O and H". This is clearly a misunderstanding.
In addition, on page 6 line 164, the authors wrote: ’this suggests that during sonication, one of the C=O bonds is destroyed.’
I agree with the authors, but maybe this bond was due to contaminants and doesn't have a clear role in the production of graphene.
The authors wrongly stated (line 152): "The XPS spectrum of Shungite does not contain any elements other than C, O and H": I've never seen an XPS spcectrum containing H spectral line. The only way to hypothesize the presence of H is because of chemical shift.
In my opinion, the manuscript must contains an XPS spectrum, and not only tables.
They are focusing their attention just on C1s spectral line, but the O1s line should also be investigated: this is why an image of the spectra should avoid misunderstanding, if the authors are firm on their position.
3) Regarding my comment on FWHM of peaks, the authors replied: "Looking through articles related to the processing of graphene oxide and graphene, we can see how the (FWHM) changes." This references should be reported.
4) Furthermore, I've asked the authors to add quantitative analysis of the relative intensities of C 1s and O 1s related components, but they didn't reply to this issue.
Author Response

(The authors gave the same response as above.)

Reviewer 2 Report
Accept in present form。
Author Response
Good day, please see the attachment.